# Research on a Ship Mooring Motion Suppression Method Based on an Intelligent Active Anti-Roll Platform

Feng Gao [1,2,*], Yougang Tang [2], Chuanqi Hu [1] and Xiaolei Xie [3]

1   National Engineering Laboratory of Port Hydraulic Structure Technology, Tianjin Research Institute of Water Transport Engineering, M.O.T., Tianjin 300456, China; pk80909677@163.com
2   School of Civil Engineering and Architecture, Tianjin University, Tianjin 300072, China; ejodo1999029@163.com
3   Department of Port Coastal and Offshore Engineering, Hohai University, Nanjing 210098, China; x327115436@163.com
*   Correspondence: gaofeng@tiwte.ac.cn

**Abstract:** Conventional ship mooring in ports has many shortcomings such as a high safety risk, low efficiency and high labor intensity. In order to explore and develop the theory and key technologies of intelligent automatic mooring systems, this paper takes an intelligent mooring system based on a parallel anti-rolling mechanism as the research and development object. A new mooring method integrating ship hydrodynamics, mechanism kinematics and intelligent algorithms is proposed. Through numerical simulation and comparative analysis of the model, the motion inhibition effect of mooring ships under different working conditions is obtained. The results show that the control strategy and intelligent algorithm of the system can realize the active control of the wharf mooring ships and achieve the goal of improving wharf stability conditions through an intelligent mooring system.

**Keywords:** mooring; intelligent; parallel platform; exercise; the berthing conditions





## 1. Introduction

With the development of ports, the requirements for their safe operation and their operational efficiency have constantly increased; however, the traditional operation mode of ship mooring at wharfs is facing new challenges. Unmanned wharfs have become a positive development trend in the ship and port industry in marine transportation businesses, and their technologies are being continually developed due to advancements in cable materials and ship automation [1–4]. However, as an important link between ships and shores, mooring operation methods have not made great progress [5–7]. With the growth of the global shipping market in recent years, issues such as the trend towards larger ships, increased loading and unloading effectiveness and frequent mooring operations have become commonplace in the shipping sector [8]. During the berthing period, due to the hydrometeorological conditions, loading and unloading operations caused by frequent changes in the ship's draft and other factors, it is necessary to constantly adjust the length of the cable to ensure the stable condition of the ship [9,10]—which not only increases the labor intensity on the crew, but also the risk of accidents [11]. With the continuous progress of modern science and technology, new, intelligent and unmanned port loading and unloading operation with real-time monitoring and self-adjustment has become a reality, and it has become possible to use high-end and new technology to ensure the operational safety of ships during mooring, instead of the traditional cable mode that has been used for many years. The Moor Master system developed by Cavotec SA of Switzerland, the AutoMoor system developed by Trelleborg of Sweden and the ShoreTension® of the Netherlands are all automated mooring solutions that have been put into use [12,13]. Good results have been achieved in mooring operations such as those in barge docks and locks [14,15]. At the same time, in the long run, the realization of an intelligent automatic mooring system that has the

characteristics of being safe, efficient, economical and environmentally protective—as an important aspect of intelligent ports—has led to mooring procedures increasingly paying more attention to automation; the urgency of this demand is gradually increasing and ship mooring—instead of traditional cable arrangements—will be a developing trend in the future [16]. Figure 1 shows automated mooring devices, represented by products from European companies, that have been applied.

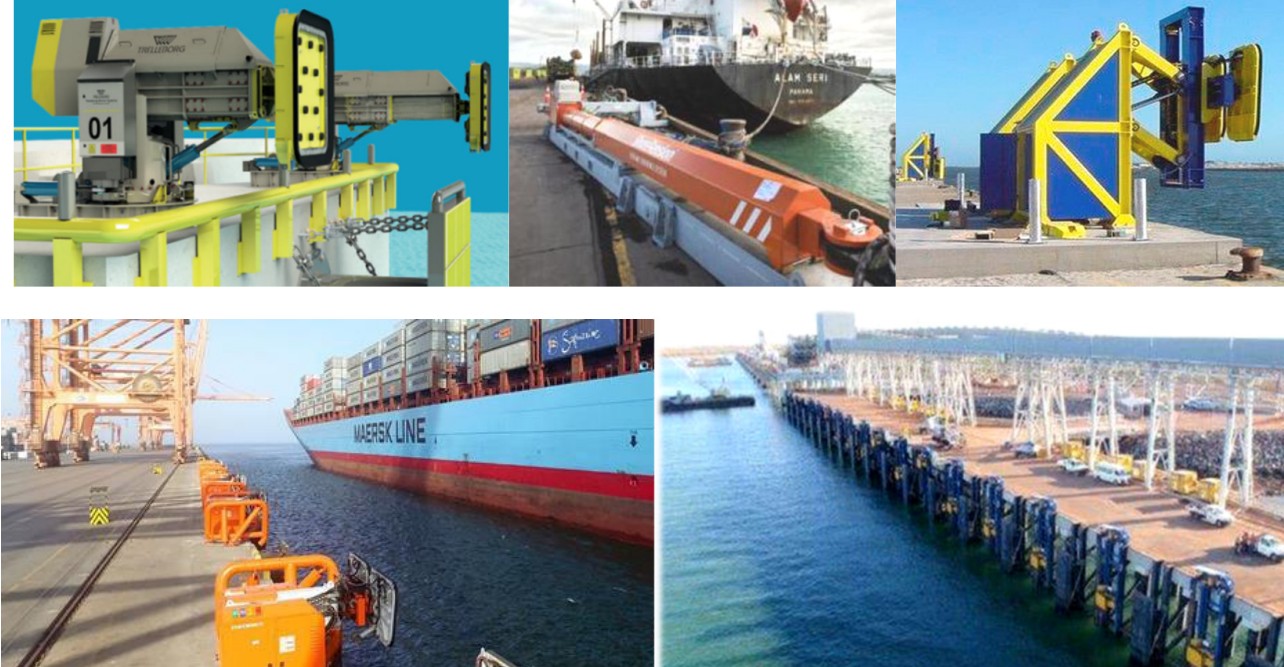

**Figure 1.** In Europe, some terminal automatic mooring equipment has been developed and applied.

In addition, concerning the theoretical research methods of intelligent mooring systems, as an automatic mooring system involves the modeling of multi-body dynamic systems controlled by fluid–structure coupling field forces and robots, their solution is very complex; thus, empirical conclusions are usually obtained by theoretical simplification, numerical simulation or model tests. However, the relevant published research data are relatively limited. De Bont et al. (2010) [17] used the MoorMaster$^{TM}$ system unit developed by Cavotec to combine various numerical models to calculate a mooring ship's motion in a harbor pool; they compared the simulation results of the MoorMaster$^{TM}$ unit's influence on the mooring ship's motion with the measured results and drew conclusions. Van Deyzen et al. (2014) [18] simulated a 3000 TEU container ship and compared the MoorMaster$^{TM}$ with traditional mooring cables and ShoreTeness@mooring systems to test whether such mooring systems can reduce the berthing time in an exposed dock. J. de Bont (2010) [19] verified the inhibition effect of MoorMaster$^{TM}$ mooring system on ship horizontal motion through numerical solution. In China, Ke (2014) [20] carried out the structural design and finite element analysis of an automatic mooring system at a port, adopted the hypothesis verification method to design its mechanism and then verified the correctness of the design through finite element analysis. Liu (2020) [21] carried out the design and analysis of an energy regenerative automatic mooring device, adopted the sliding block spring module to simulate the effect of sea waves on the generalized ship and analyzed the influence of damping on the ship's motion response. Zhao et al. (2022) [22] proposed a new robot mooring system based on an underactuated mechanism, established the global stiffness model of the mooring robot system based on the influence coefficient method and obtained the stiffness matrix in the form of the tensor.

In this paper, an intelligent mooring system based on a parallel anti-rolling mechanism is developed, and a new mooring method integrating ship hydrodynamics, mechanism kinematics and intelligent algorithms is proposed. First, in Section 2, the mooring motion equation is analyzed based on the action mechanism and load characteristics of the intelligent mooring system on ship motion, and the intelligent mooring motion equation in the time domain is obtained. In Section 3, an overall design scheme for an intelligent mooring system based on a parallel mechanism is proposed—that is, a dual platform configuration consisting of two parallel platform units (3UPS/6UPS) with six degrees of freedom of movement. In Section 4, an intelligent algorithm controller (fuzzy_control) is generated using Matlab tools and converted into a user-defined external load dynamic link library file that can interact with AQWA to simulate the action process of the intelligent mooring system in the time domain calculation of ship hydrodynamics, and then, a simulation model of the fully coupled dynamic response of the intelligent mooring system is established. Finally, in Section 5, the mooring experiments of the scaled-down verification prototype device (SVPD) are described. In addition, the conclusions are given in Section 6.

## 2. Intelligent Mooring Motion Equation in the Time Domain

### 2.1. Motion of a Mooring Ship under Wave Action

Based on the frequency-domain calculation of the three-dimensional potential flow theory, ship hydrodynamic parameters such as the amplitude operator of motion response, wave drift force, first-order wave force, additional mass and damping coefficient under wave action can be obtained. The frequency domain theory is generally only applicable to the interactions between regular waves and linear constraint systems [23]. However, in practical engineering, mooring ships are jointly constrained by mooring cables and fenders at the same time, and their forces are nonlinear. In the movement process of wharf mooring ships under the combined action of winds, waves and currents, the cables will be strained and relaxed, and the fender will be compressed and rebound. Therefore, the ship's motion is also nonlinear. Due to the difference in the stiffness of the cable and the fender, even if the two are assumed to be linear, the ship's mooring force and its motion response are also complex nonlinear problems, and the ship's motion response is no longer simply a harmonic vibration.

The six-degree-of-freedom dynamic equation of mooring ships in unidirectional irregular waves proposed by Cummins in 1962 [24]—namely, the nonlinear motion equation of mooring ships—is solved by Fourier transformation through the use of hydrodynamic coefficients such as the ocean wave excitation force, the ship's additional mass and the radiation damping calculated in the frequency domain. The wave force, added mass and hysteresis functions of floating bodies such as ships are obtained in the time domain. Finally, the numerical results of the six degrees of freedom of motion (heave, roll, surge, sway, pitch and yaw) and mooring elements such as the mooring force, fender impact force and fender compression deformation are solved according to the coupled motion equation in the time domain—see Figure 2. The expression of the time domain analysis is as follows:

$$(M + m)X''(t) + \int_0^\infty X'(t)R(t - \tau)d\tau + KX(t) = F^{(1)} + F^{(2)} \tag{1}$$

where $X(t)$ is the motion vector of the mooring ship—namely, the motion with six degrees of freedom: pitch, roll, heave, roll, pitch and yaw; $M$ is the hull mass inertia matrix; $m$ is the fluid added mass matrix; $K$ is the still water restoring force coefficient matrix of the hull; $F^{(1)}$ is the first-order wave excited force vector on the hull; $F^{(2)}$ is the second order wave excited force vector of the hull; $R(t-\tau)$ is the hull velocity pulse function matrix and the convolution integral term represents the fluid memory effect.

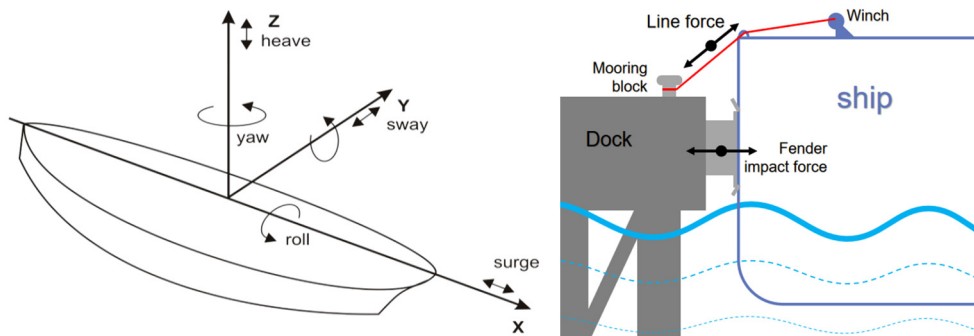

**Figure 2.** Motion and force diagram of a mooring ship.

In the above equation, for the added mass, m can be obtained as follows:

$$m = A(\infty) \tag{2}$$

where $A$ is the frequency-dependent additional mass matrix composed of $A_{jk}$ and $A_{jk}$ is the KTH additional mass coefficient generated by the rigid body motion of the JTH unit amplitude body, where $j = 1,2 \ldots N$; $N$ is the degree of freedom.

The function $R(t)$ can be obtained by the inverse Fourier transform of the frequency-dependent radiation damping coefficient, expressed as follows:

$$R(t) = \frac{2}{\pi} \int_0^\infty B(\omega) \cos(\omega t) dt \tag{3}$$

where $B$ is a frequency-dependent potential damping matrix composed of $B_{jk}$ and $B_{jk}$ is the KTH potential damping coefficient generated by the JTH unit amplitude rigid body motion.

### 2.2. Time Domain Motion Mechanics Analysis Based on Intelligent Mooring

Since the force vector of the conventional ship mooring time domain equation includes the contribution of waves, winds, waves, currents, mooring cables and fenders, and the mooring motion equation is solved in the time domain, the nonlinear characteristics of mooring cables and fenders can be directly considered. Similarly, the complex characteristics of the influence of each unit of the intelligent mooring system on the ship's motion response can be considered in the above time-domain equation. Therefore, if the force F of the intelligent mooring system on the ship is added to Equation (1) above, the exciting force generated by the mooring system instead of the traditional cable and fender can be reflected. Then, the equation above—after considering the influence of the force generated by the intelligent mooring system device—can be expressed as:

$$(M + m)X''(t) + \int_0^\infty X'(t)R(t - \tau)d\tau + KX(t) = F^{(1)} + F^{(2)} + F \tag{4}$$

$$F = \sum_{q=1}^n F_q \tag{5}$$

In the above equation, $F$ is the mooring force exerted on the ship by the intelligent mooring system. In the active control mode of the intelligent mooring system, the external load is mainly generated by the mooring system mechanism. In actual system operation, the mooring force can be the resultant force of multiple intelligent mooring units; $F_q$ is the force provided by the QTH intelligent mooring system and $n$ is the number of mooring system units, which can be understood as the number of mooring devices.

Therefore, if adopting an intelligent mooring system instead of cables and fenders, it is then necessary to find a suitable mooring force *F* to effectively inhibit the motion of the mooring ship. Obviously, the intelligent mooring system will be realized mainly through the mechanical friction in the mooring mechanism itself and the damping of the actuator.

## 3. Mechanism Design Based on Multi-Body Dynamics

### 3.1. Overall Structure of the System

The mooring mechanism needs to meet the requirements of six degrees of freedom, and also needs to have good impact resistance and high stiffness characteristics. Therefore, it is necessary to rely on a stable and reliable anti-roll platform device. Therefore, by referring to the characteristics of the classic Stewart mechanism and its derivative mechanism in the parallel mechanism [25,26], in combination with the basic requirements of ship mooring, the double parallel configuration is adopted—namely, a six drive, three power structure (referred to as 6UPS/3UPS). The double parallel mechanism consists of six driving branches, 12 power branches and upper and lower platforms. The branch is connected to the lower platform through a hook hinge, the upper platform through a ball hinge, and the mobile pair drive the branch as the driving pair. The mobile assistant of the auxiliary branch acts as an auxiliary power [27,28].

A double parallel berthing mechanism can be separated into the independent control stages of the single parallel units (i.e., before contact with the ship model and after separation from the ship model) and synchronous control stage of the double parallel unit (i.e., both double parallel units are adsorbed within the ship model). The two lower platforms of the double-parallel docking mechanism are fixed with the shore, which can be regarded as a large plane fixedly connected to the shore. When the upper platforms of two parallel units are in contact with the sideboard of the ship model, the two upper platforms are coplanar, which can also be equivalent to a large plane being in contact with the ship. A schematic diagram of double parallel mechanism and an equivalent model of a mooring system constituted by it are shown in Figure 3.

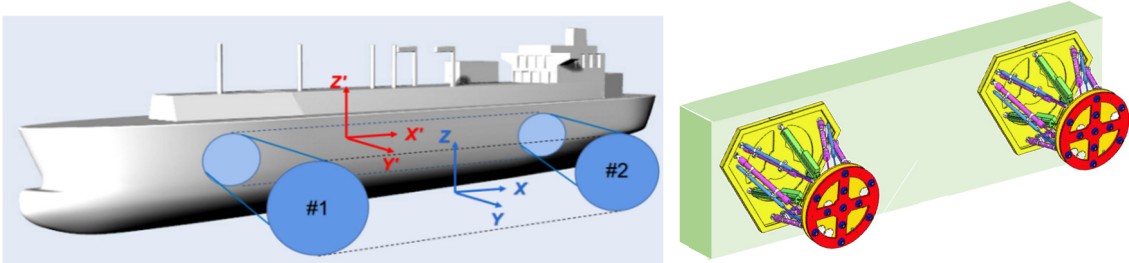

**Figure 3.** Equivalent model schematic diagram of a double parallel butt mechanism.

### 3.2. Control Strategy

In traditional dock mooring operations, cables and fenders provide elastic restoring forces and damping forces for the entire ship–shore mooring system to restrict the ship's movement within a certain range, while traditional cables and fenders provide a mooring stiffness for the ship. When the ship is subjected to hydrodynamic forces such as waves and currents, the entire mooring system can resist the ship's movement in the form of the elastic restoring force—just like the role of force vector F in the analysis of the ship motion time-domain equation mentioned above. If the intelligent automatic mooring system is used to operate the ship instead of the traditional cable and fender device, the intelligent mooring system will provide the ability to resist changes in terminal position and pose when the ship is subjected to external forces such as waves, and also play the role of force vector *F*. Through the analysis of the motion and force characteristics of the ship mooring system mentioned above, it can be found that the existence of both the restoring force and damping force can restrain the slow drift of the mooring ship and are more conducive to reducing the low-frequency amplitude of the mooring ship. Based on this, the active

control of an intelligent automatic mooring system can be realized through the following three methods:

(1) Friction damping is provided by auxiliary branches in the parallel system, which is a passive force and cannot be controlled by the intelligent mooring system;

(2) The aerodynamic damping provided by the driving branch in the parallel system is the active force, which can be interfered with by adjusting the relevant parameters of the intelligent mooring system. The damping force can inhibit the amplitude of the ship at various frequencies;

(3) The constant pressure at both ends of the driving branch is maintained to provide the restoring force, which is the active force, and can be intervened by adjusting relevant parameters of the intelligent mooring system. The restoring force can change the slow drift motion of the ship into a low frequency oscillation.

Figure 4 shows the control strategy diagram of the intelligent mooring system. According to the requirements of the mooring control task, it is necessary for the intelligent mooring system to maintain the ship's attitude within the range of motion expected in safe mooring standards. Therefore, by means of the mooring force—composed of the damping force and restoring force—the position and posture of the adsorbed ship can be adjusted actively. Among these, the damping force can suppress the ship's amplitude at various frequencies, and the restoring force can change the ship's slow drift motion into a low frequency oscillation. What the intelligent mooring system can control is the damping $F_{2i}$ provided by the drive branch and the restoring force $F_{1i}$ provided by the constant pressure at both ends of the drive branch. Therefore, in order to make the control strategy of the above system feasible, the damping coefficient and restoring force coefficient of the control drive branch are largely regulated, so as to realize the active control of the mooring power output equipment—the drive branch.

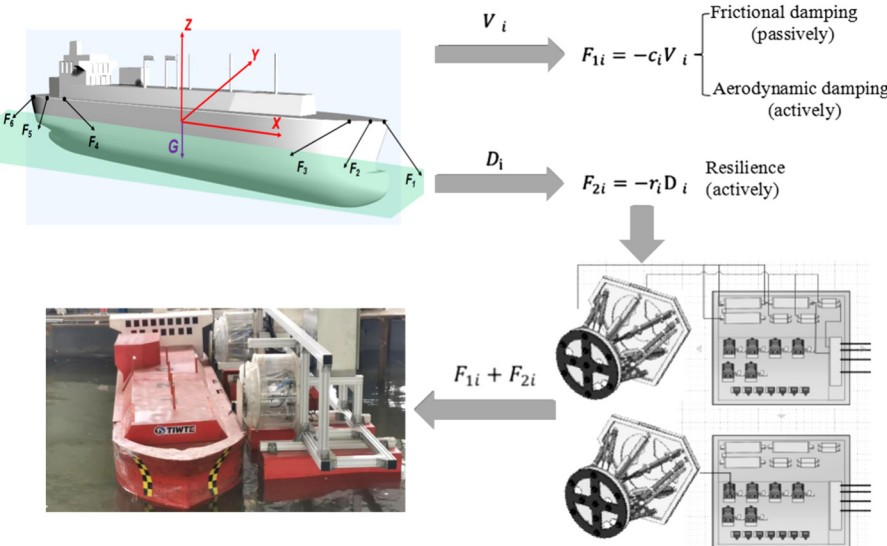

**Figure 4.** Control strategy diagram of an intelligent mooring system.

Considering that intelligent mooring systems are highly nonlinear, strongly coupled complex control objects, in order to obtain good control accuracy and robustness—and to meet the requirements of ship-to-shore environment engineering availability under wave and current conditions—it is necessary to adopt an intelligent control algorithm with low computational complexity, good control performance and relative maturity; this will be studied and analyzed in the next chapter.

## 4. Intelligent Algorithm and Simulation Modeling

### 4.1. Intelligent Control Algorithm

Fuzzy control is an intelligent control algorithm based on fuzzy mathematics. It is a new control method based on fuzzy set theory and fuzzy logic reasoning and sensor technology and computers. This algorithm is a controller designed on the basis of human control experience of the controlled object; it does not depend on the precise mathematical model of the controlled object and can realize effective control of complex uncertain nonlinear systems. It is usually applied to robot controls. In this paper, the Fuzzy Logic toolbox in Matlab is integrated with the Fuzzy Inference System (FIS) editor, the membership function editor, the fuzzy rule editor, the rule browser and output preview and other visual tools, which facilitate the construction of fuzzy control rules quickly, avoiding complex programming process.

The objective of intelligent mooring control is to limit the movement of mooring ships near the set balance position. In order to facilitate this setting in FIS (see Figure 5), the deviation between the actual movement of ships and the value of the balance position is used as the control rule to set relevant parameters. Among them, there are two input quantities: the SD represents the deviation of exercise quantity and the rate represents the variation of its deviation. The output quantity is the mooring coefficient Damp or Resilience of the intelligent system. In this way, a two-dimensional fuzzy controller is constituted—namely, two inputs and one output. Fuzzy control consists of a series of "If-Then" statements, and its operating experience is as follows: when the ship exceeds the balance position of the movement set or tends towards doing so, the mooring coefficient is increased; the mooring coefficient is reduced when the ship approaches the motion equilibrium position or tends towards doing so.

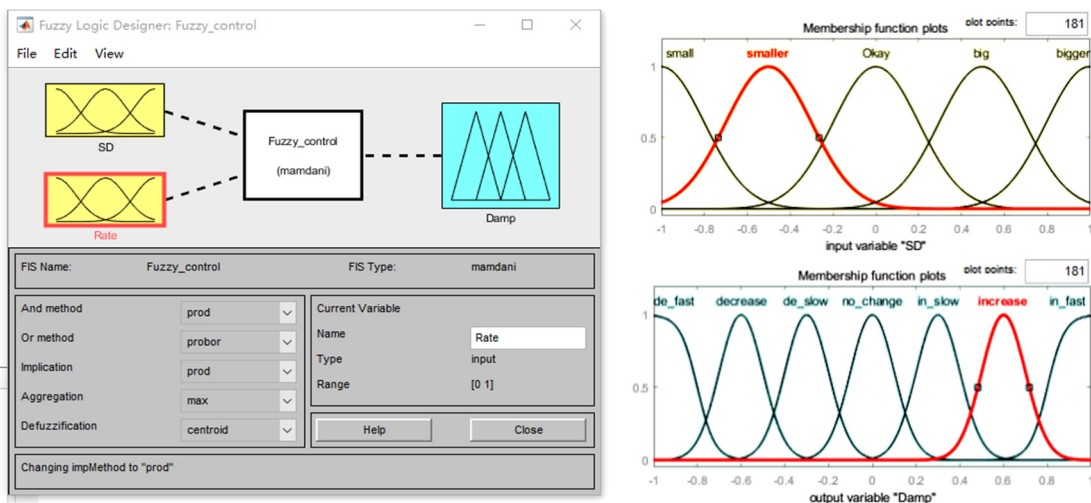

**Figure 5.** The input variable, output and its fuzzy rules can be set through the FIS editing interface.

In specific file Settings—for example, the SD variable—the language for setting the five basic variables is as follows: (1) If SD is okay then damp is not changed; (2) If SD is big then damp is increased; (3) If SD is small then damp is decreased; (4) If SD is close to okay and rate is positive then damp is increased slowly; (5) If SD is close to okay and rate is negative then damp is increased quickly.

The above five rules are the most basic form of control rules, and they can form a rulebase consisting of $5 \times 5 = 25$ rules. If this is further refined on the basis of the above five, more perfect and fine fuzzy reasoning rules can be formed. If there are seven variables, $7 \times 7 = 49$ rules can be reached.

Through the reasoning process of the fuzzy rules mentioned above, and then through centroid defuzzification, the clear output of the continuous quantity of fuzzy output can be obtained—that is, the mooring coefficient at the current control time. The program will be

saved as the intelligent controller file with the suffix "*.fis". This file is compiled into a DLL file (fuzzy_contro.dll) for interactive use in the subsequent ship hydrodynamic simulation modeling.

### 4.2. Modeling and Simulation Verification

ANSYS AQWA has the function of imposing User Force or External Force through a built-in dynamic link library [29,30], which can realize the purpose of calling an intelligent mooring system control program in ship hydrodynamic calculations. At this time, the simulation of the intelligent mooring function can be regarded as realizing the effect of a damping system, with special properties between the ship and the dock through this User Force.

In a complete simulation with a duration of T and a time step of $\Delta t$, AQWA is used to calculate the hydrodynamics of mooring ships, and the amount of ship exercise is read in real time from the AQWA. The mooring response of the intelligent mooring system is completed through the fuzzy logic controller dynamic link library file (DLL file) established in the previous section of this paper, and fed back to the mooring ship in the AQWA to realize the response to the current moment (*t*); the result is used to respond to the interaction influence of the next time step (*t+*). The intelligent control program (DLL file) and AQWA continue to repeat the above process and exchange data until the simulation is successfully completed. The time domain analysis process of the above combined simulation is shown in Figure 6 below.

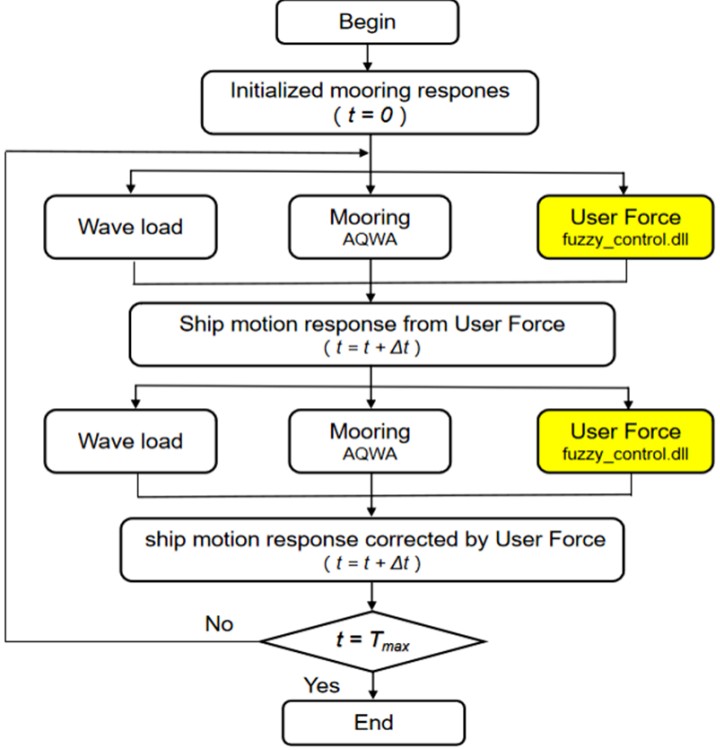

**Figure 6.** Flow chart of the numerical simulation. (The yellow highlight shows the location of fuzzy logic controller).

In this study, an LNG carrier was used to conduct the AWQA hydrodynamic simulation. The main scale of the ship type is shown in Table 1, and the ship hydrodynamic grid model is shown in Figure 7. In the simulation calculation, the port side of the ship was set to dock beside the wharf arranged along the shore, and the starboard side was set to survey the sea. As there was no fender, the distance between the hull and the wharf was set at about 3.0 m to reserve movement space. The time step in the AQWA was set to 0.1 s, and the number of grids calculated was 3087.

**Table 1.** Characteristics of the moored ship.

| Designation | Ship | Unit |
|---|---|---|
| LOA. | 315 | m |
| LPP. | 290 | m |
| Breath | 50 | m |
| Depth | 27 | m |
| Draft | 12.0 | m |
| Displacement | 146,871.2 | t |
| Roll period | 14.92 | s |

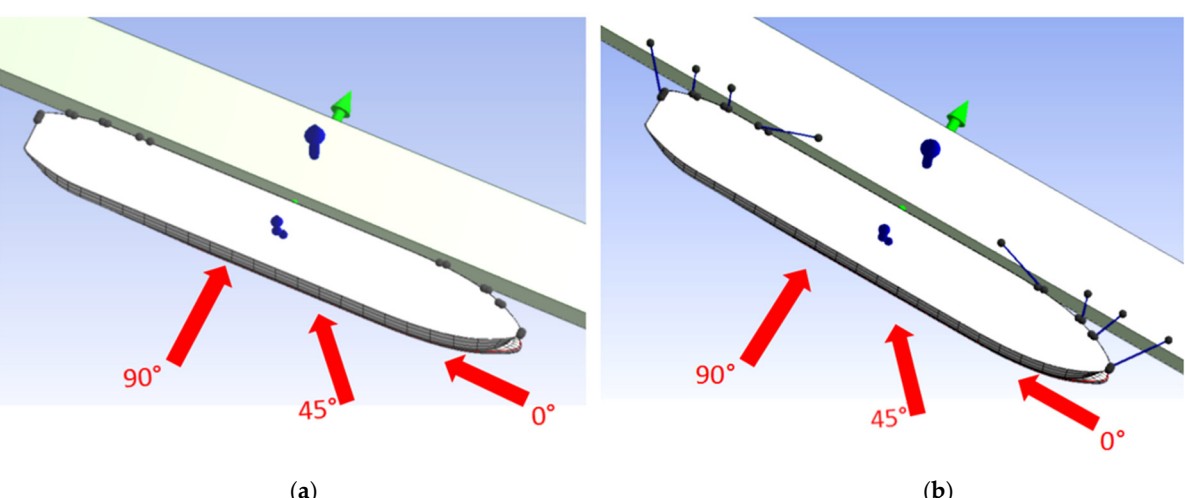

(**a**)   (**b**)

**Figure 7.** Comparison calculation model of the two mooring modes. (**a**) Intelligent mooring arrangement; (**b**) Conventional mooring arrangement.

In order to facilitate the comparison effect, the simulation calculation of the conventional mooring mode was also carried out in the simulation. The mooring arrangement mode was 3:3:2:2, the cable uses HMPE, and the fender was an SUC2500H-R0 rubber fender. The specific parameter-setting process of the AQWA is not described here. The calculated water depth, waves and other environmental conditions in the model were consistent with those in the intelligent mooring simulation.

*4.3. Analysis of the Simulation Test Results*

It can be seen from the simulation results for the exercise amount (see Figure 8) that the motion inhibition effect of intelligent mooring and conventional mooring was similar, and roll was more obvious in the overall performance of each movement. By comparing the results of the two mooring modes, the active control of the mooring damping force achieved by the intelligent algorithm was able to achieve the purpose of reducing the ship's motion amplitude. Through the setting and judgment of the motion threshold in the intelligent control, the control external force of the damping force set was able to provide the ability to restrain the amplitude of the ship at various frequencies, making the ship's motion amplitude oscillate within the expected range. It can be seen from the change trend of the approximate amount of exercise under the two mooring modes that the intelligent mooring mode was able to achieve a motion inhibition effect similar to the conventional mooring mode in terms of the mooring constraints, and even slightly better than the conventional mooring mode in some exercise performance parameters.

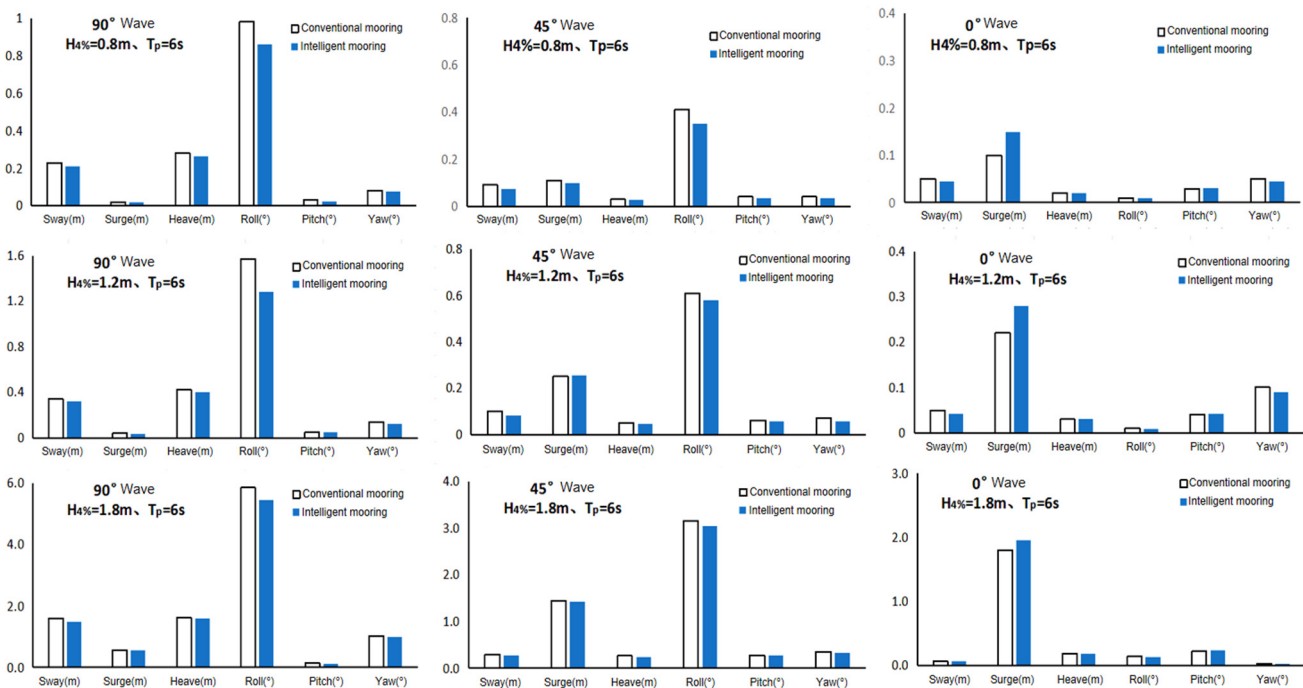

**Figure 8.** Comparison of the calculation results for ship movement under the two mooring modes.

Under the same working conditions, both the intelligent mooring system and the conventional mooring arrangement were able to achieve the control effect of the wharf mooring ship. From the comparison of the simulation results, the range of motion under intelligent mooring control was reduced, the average amplitude of each operating condition was about 10% and the effect of transverse wave was relatively more obvious. The results showed that the automatic system can indeed play a more advantageous role. Meanwhile, with increases in wave height, the improvement effect gradually decreased. For example, the improvement effect was more significant when $H_{4\%}$ = 0.8 m than when $H_{4\%}$ = 1.8 m.

## 5. Scale Verification Machine Model Test

### 5.1. About SVPD

On the basis of the above research, after determining the key components and parameters, a set of intelligent mooring scaled-down verification prototype devices (SVPDs) containing two parallel mechanism units were developed and produced, and the ship model test was carried out in a wave pool to verify the feasibility of the intelligent mooring system design scheme and the effectiveness of the intelligent algorithm.

An SVPD is an electromechanical and pneumatic integration system based on a double-parallel six-degree-of-freedom mechanism configuration, which is mainly composed of a mechanical body, sensing and detection part, pneumatic system and electronic control system. The double parallel mechanism was numbered #1 unit and #2 unit, respectively, and the carrying capacity of each unit was proposed to be 100 N. In terms of the overall arrangement, along the horizontal direction, it could be adjusted along the track according to the requirements of the spacing of mooring points; along the vertical direction, it could be adjusted along the track according to the water level, the depth of the ship type and the draft. The lower platform of the SVM and pier were fixed through an L-shaped connection frame and the upper platform was in contact with the test ship model through the adsorption unit, as shown in Figure 9.

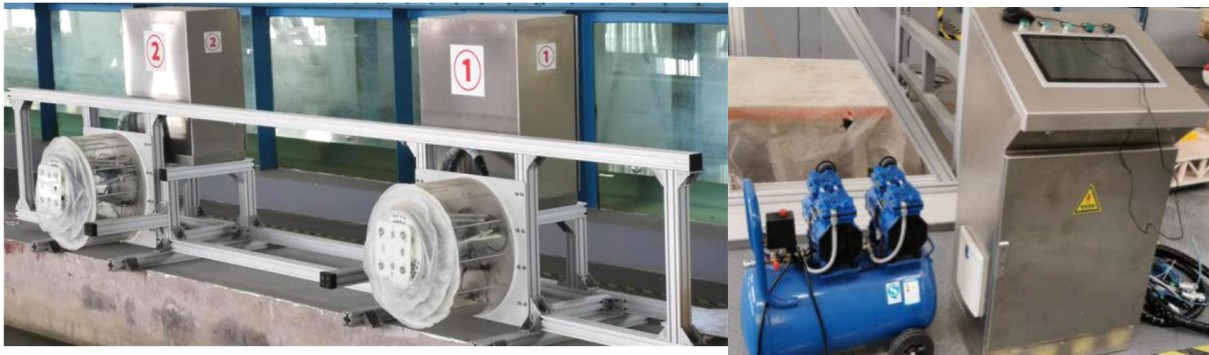

**Figure 9.** Intelligent mooring scaled-down verification prototype device (SVPD).

*5.2. Design of the Test Model*

According to the size of the created intelligent mooring prototype, the ship model with a consistent scale was matched and the effect was tested in the wave test pool. The target ship type was an LNG ship type ($\lambda = 60$). Considering that sufficient freeboard height should have been left to meet the adsorption plate of the device, the loading state of the ship model was only one state of ballast. The design of the ship model based on gravity similarity needed to satisfy geometric similarity, static similarity and dynamic similarity. The center of gravity and roll and pitch period of the ship were calibrated and verified to satisfy the conditions of similarity. Figure 6 shows the Wave test tank and test ship model.

The model plane layout of the test is shown in Figure 10. The test basin contained a test ship model, a pier and a platform for placing the intelligent mooring system control host.

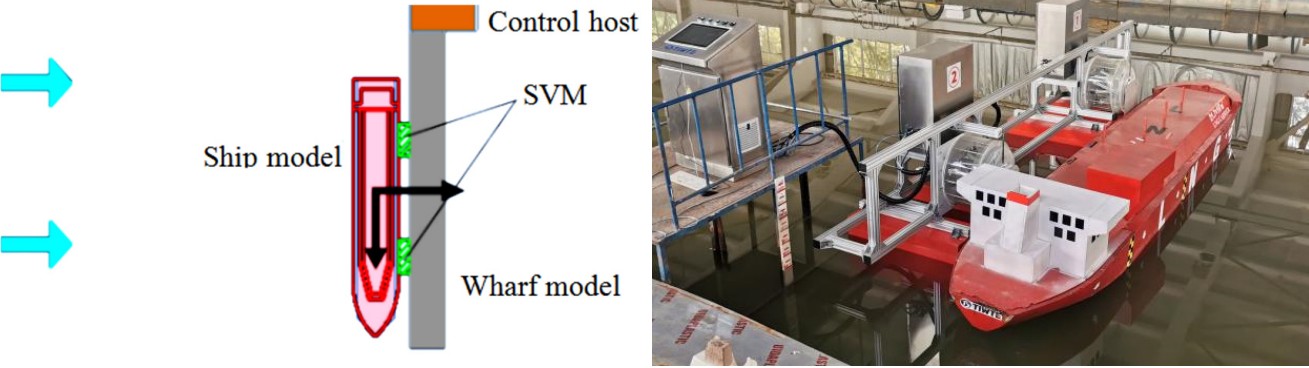

**Figure 10.** SVPD test model plan in the wave test basin.

*5.3. Analysis of the Test Results*

Figures 11 and 12, respectively, show the process of ship movement when the intelligent system was working (SVPD On)/not working (SVPD Off) under the action of a 90° transverse wave, different wave height and period.

The test results show that the ship's movement under the action of irregular waves could be inhibited even after the intervention of intelligent mooring system. Through the statistical analysis of the numerical results of the two types of ship's maximum movement and their standard deviation—whether from the maximum of the six degrees of freedom or its standard deviation—the value of the intelligent automatic mooring system during operation was smaller than the result when the ship's movement was not running, and the suppression effect on the ship's movement was reflected. Relative to the angle, the horizontal displacement was improved more obviously, and the most significant reduction in the movement amplitude appeared in rolling.

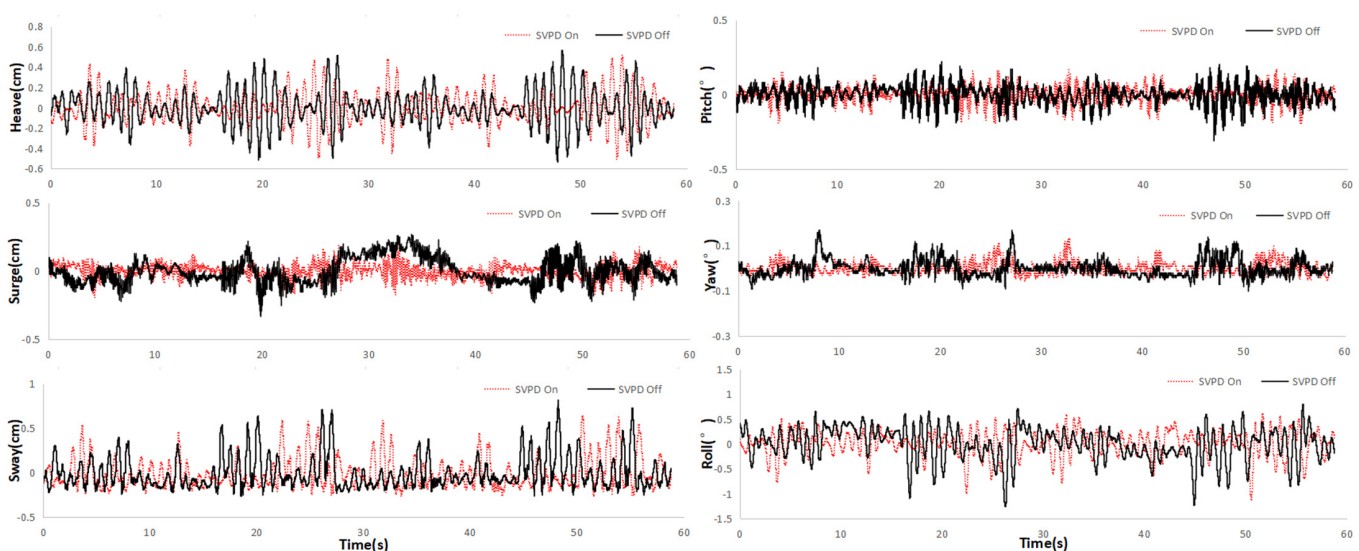

**Figure 11.** Test results of ship model movement before and after the intervention of intelligent mooring system. ($H_s$ = 2.4 cm, T = 0.8 s, irregular waves).

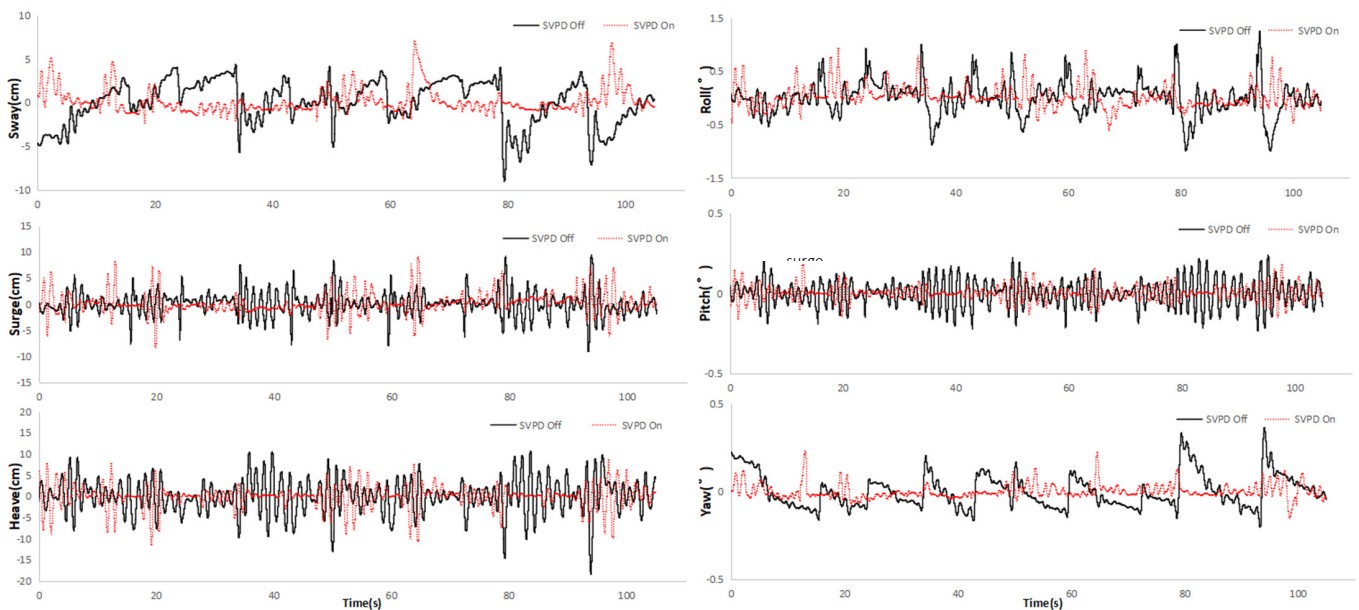

**Figure 12.** Test results of ship model movement before and after the intervention of the intelligent mooring system ($H_s$ = 3.3 cm, T = 0.8 s irregular waves).

## 6. Conclusions

This paper proposed a scheme for an intelligent automatic ship mooring system based on a double parallel mechanism, which can be used to restrain the motion of mooring ships. This system can automatically adjust the mooring force using a fuzzy logic algorithm, realize the active control of a ship's movement and achieve intelligent mooring operations at a wharf. The ship motion inhibition effect when the moored ship encountered waves was verified, respectively, by simulations and model tests. The results showed that the control strategy and intelligent algorithm were able to effectively suppress the movement of the ship with six degrees of freedom. SVPD can provide the advantage of active intervention to achieve a more intelligent multi-dimensional motion inhibition. Considering the limitations of model testing and the perfection of the intelligent mooring theory, there is still a certain gap in applicability to actual ports and terminals. The research in this paper establishes the feasibility of an intelligent system based on parallel mechanisms in mooring operations,

which provides support for subsequent studies on intelligent mooring control prototype design and related theory.

**Author Contributions:** Conceptualization, F.G.; Methodology, F.G.; Software, Y.T.; Validation, X.X.; Formal analysis, F.G. and C.H.; Data curation, X.X.; Writing—original draft, F.G.; Writing—review & editing, Y.T.; Funding acquisition, F.G. All authors have read and agreed to the published version of the manuscript.

**Funding:** The authors would like to acknowledge the support of the National Natural Resources Fund (Grant No. U21A20123) and the Central Commonweal Research Institute Basic R&D Special Foundation of TIWTE, China (Grant No. TKS20200304).

**Data Availability Statement:** The data used to support the findings of this study are included within the article.

**Conflicts of Interest:** The authors declare that they have no competing interest.

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
