# Peer review of "Research on a Ship Mooring Motion Suppression Method Based on an Intelligent Active Anti-Roll Platform"

_applsci, doi:10.3390/app13137979_

Round 1
Reviewer 1 Report
The peer-reviewed manuscript presents an intelligent mooring system based on parallel anti-rolling mechanism. The problem under consideration is very topical and relevant. The discussed issue is supported by citations of current scientific papers. The paper presents theoretical considerations, results of numerical analyzes and results of a laboratory experiment. The manuscript is well written and surprisingly concise. Some minor remarks are presented below.
1. I propose to change the form of the caption under figure 1. Currently, the caption under the figure speaks of a certain action (have been applied), which is quite unusual.
2. Do not use a colon after section titles (even if it is an Introduction or Conclusions).
3. Page 3 under equation (1) "six degrees of freedom: pitch, roll, heave, roll, pitch and yaw". I suggest explaining these degrees of freedom in a figure or refer to a figure in the manuscript (possibly Figure 3).
4. In Fig. 3, the symbols V and D are shown, although it is not explained exactly what they mean.
5. Page 7 under Fig. 4 "The language for setting the five basic variables is as follows: ...". I don't understand the rest of the sentence. There are numerous repetitions, although I do not know if they are intentional, and if so, why.
6. Page 9 "Compared with the conventional mooring mode, it can achieve similar effects to the conventional mooring arrangement conditions in terms of ship mooring constraints, and even slightly better than the traditional mooring mode in terms of some exercise performance." This sentence is not fully understood. I don't know what the intention of the authors was.
7. Figures 10 and 11 are unreadable. The description of the results presented in these figures also seems quite poor.
Reviewer 2 Report
The contents are good and it could further being strengtened. Here are the comments:
-
Figure 1: Could you please provide a reference for this figure? Additionally, the labeling on the figure seems to explain the different purposes. For instance, the ship hydrodynamic grid model is shown in Figure 6. Please clarify this statement.
-
It would be helpful to provide a detailed explanation for the change in degree from the conventional mooring arrangement (90-60-0) to 90-45-0 in the intelligent mooring arrangement. Justification or analysis is needed to support this decision.
-
Figure 7: Could you please explain the calculations that resulted in this figure? It would enhance the understanding of the presented data.
-
The paper mentions three methods for active control: friction damping, aerodynamic damping, and constant pressure. It would greatly benefit the reader to include a comprehensive discussion that explains these methods in detail. Additionally, specific information regarding the friction parameters, relevant parameters, pressure amount, etc., should be provided.
-
It is essential to explain the limitations of the model and the assumptions made for the test model. Since the test model differs from real-world situations, it is important to provide insights to the readers regarding potential considerations in practical applications. For example, discussing infrastructure requirements, conducting a cost/financial analysis comparing the conventional and proposed technology, and addressing maintenance issues would add value to the paper.
-
Figure 11: The results of the ship model's movement need to be comprehensively explained. What are the main findings, and how do these graphs differ from one another?
-
It appears that there is a discrepancy in the terminology used. The author mentions the use of a double parallel mechanism as the approach, which differs from what has been mentioned in the title and the main content of the manuscript. Please double-check and clarify this terminology.
-
As this paper is submitted to the Applied Sciences journal, it would be appropriate to include references from this journal to support the research and align it with the field of study.
The English language used in the manuscript is generally understandable, but there are several areas where improvements can be made to enhance clarity and coherence. Here are some observations regarding the quality of the English language:
-
Sentence structure: Some sentences are long and convoluted, making it difficult to grasp the intended meaning. Breaking them down into shorter sentences or using appropriate punctuation would improve readability.
-
Grammar and punctuation: There are instances where proper grammar and punctuation could be improved. For example, there are missing commas in compound sentences and incorrect capitalization in some instances.
-
Verb tense consistency: There are shifts in verb tenses within sentences and between paragraphs. Maintaining consistent verb tenses throughout the content would enhance clarity.
-
Word choice and phrasing: Some phrases and word choices could be more precise and concise. Simplifying complex sentence structures and using plain language would make the content more accessible to a wider audience.
